# Programming Mechanism of Adipose Tissue Expansion in the Rat Offspring of Obese Mothers Occurs in a Sex-Specific Manner

**DOI:** 10.3390/nu15102245

**Published:** 2023-05-09

**Authors:** Carlos A. Ibáñez, Gabriela Lira-León, Luis A. Reyes-Castro, Guadalupe L. Rodríguez-González, Consuelo Lomas-Soria, Alejandra Hernández-Rojas, Eyerahí Bravo-Flores, Juan Mario Solis-Paredes, Guadalupe Estrada-Gutierrez, Elena Zambrano

**Affiliations:** 1Departamento de Biología de la Reproducción, Instituto Nacional de Ciencias Médicas y Nutrición Salvador Zubirán, Mexico City 14080, Mexico; carlos_albertoibc@comunidad.unam.mx (C.A.I.); gabsslln@gmail.com (G.L.-L.); luis.reyesc@incmnsz.mx (L.A.R.-C.); guadalupe.rodriguezg@incmnsz.mx (G.L.R.-G.); mconsuelo@conacyt.mx (C.L.-S.); alehero03@gmail.com (A.H.-R.); 2CONACyT-Cátedras, Investigador por México, Departamento de Biología de la Reproducción, Instituto Nacional de Ciencias Médicas y Nutrición SZ, Mexico City 14080, Mexico; 3Departamento de Inmunobioquímica, Instituto Nacional de Perinatología Isidro Espinosa de los Reyes, Mexico City 11000, Mexico; eyerahiqfb@yahoo.com.mx; 4Departamento de Investigación en Salud Reproductiva y Perinatal, Instituto Nacional de Perinatología Isidro Espinosa de los Reyes, Mexico City 11000, Mexico; juan.solis@inper.gob.mx; 5Dirección de Investigación, Instituto Nacional de Perinatología Isidro Espinosa de los Reyes, Mexico City 11000, Mexico; guadalupe.estrada@inper.gob.mx

**Keywords:** maternal obesity, adipocyte size distribution, sexual dimorphism, adipose tissue, adipogenic capacity, developmental programming

## Abstract

We investigated whether excessive retroperitoneal adipose tissue (AT) expansion programmed by maternal obesity (MO) affects adipocyte size distribution and gene expression in relation to adipocyte proliferation and differentiation in male and female offspring (F1) from control (F1C) and obese (F1MO) mothers. Female Wistar rats (F0) ate a control or high-fat diet from weaning through pregnancy and lactation. F1 were weaned onto a control diet and euthanized at 110 postnatal days. Fat depots were weighed to estimate the total AT. Serum glucose, triglyceride, leptin, insulin, and the insulin resistance index (HOMA-IR) were determined. Adipocyte size and adipogenic gene expression were examined in retroperitoneal fat. Body weight, retroperitoneal AT and adipogenesis differed between male and female F1Cs. Retroperitoneal AT, glucose, triglyceride, insulin, HOMA-IR and leptin were higher in male and female F1MO vs. F1C. Small adipocytes were reduced in F1MO females and absent in F1MO males; large adipocytes were increased in F1MO males and females vs. F1C. Wnt, PI3K-Akt, and insulin signaling pathways in F1MO males and *Egr2* in F1MO females were downregulated vs. F1C. MO induced metabolic dysfunction in F1 through different sex dimorphism mechanisms, including the decreased expression of pro-adipogenic genes and reduced insulin signaling in males and lipid mobilization-related genes in females.

## 1. Introduction

Epidemiological and experimental studies have demonstrated that maternal under- and over-nutrition during development predispose offspring to long-term adverse outcomes in a sex-dependent manner [1]. Specifically, maternal obesity leads to excessive adipose tissue accumulation in the offspring. We [2,3,4] and others [5,6,7] demonstrated that maternal obesity programs their offspring to gender-specific metabolic alterations. The expansion pattern of white adipose tissue (WAT) is strongly linked to a greater possibility of developing metabolic alterations. The sex-specific metabolic alterations in programmed obesity is driven by an altered adipose tissue expansion mechanism. Therefore, elucidating the molecular mechanisms by which maternal obesity programs offspring’s adipose tissue expansion is an important issue of physiological and clinical relevance in the context of the global obesity epidemic [8]. However, the molecular mechanisms underlying maternal obesity-related increases in adipose mass, including several aspects of the physiological development of adipose tissue in vivo, have yet to be elucidated [9,10].

Adipose tissue expansion, namely adipogenesis, occurs via the recruitment of adipocyte precursors, which become newly differentiated adipocytes via increasing the size of existing adipocytes [11]. The recruitment of plastic adipocyte precursors is an active process in the fetal rat within the last week of gestation and is accelerated during lactation [12], with both periods considered windows of vulnerability for developmental programming. However, the WAT retains a certain expansion capacity during postnatal and adult life, dependent upon sex, energy balance and perinatal conditions [13]

In rats, the expansion rate of fat depots differs according to their location. Inguinal fat, for instance, exhibits a high degree of hyperplasia, whereas mesenteric and epididymal fat depots exhibit a high degree of hypertrophy. In the present study, retroperitoneal fat tissue was chosen for examination because it is expanded by a balance of hyperplasia and hypertrophy, and its growth is associated with metabolic alterations [13,14,15,16]; also, it is one of the most sensitive fat depots to sex-specific developmental programming by maternal obesity [6]. Retroperitoneal adipose tissue expansion is regulated by several transcription factors, resulting in an increase in adipocyte size (AS) and/or number, and is linked with insulin resistance and dyslipidemia [17]. Evaluating changes in AS and/or adipocyte number and the expression profile of adipogenesis-related genes is a widely accepted method to assess changes in WAT expansion [11].

Considering that the abnormal distribution of AS and its deleterious effects on metabolic function are different between the male and female offspring of obese mothers, we hypothesized that gene transcription profiles involved with retroperitoneal adipocyte proliferation and differentiation could be sex-specifically regulated.

## 2. Methods

### 2.1. Animals

All procedures were approved by the Animal Experimentation Ethics Committee of the Instituto Nacional de Ciencias Médicas y Nutrición Salvador Zubirán (INCMNSZ), Mexico City, Mexico (ethical approval code, BRE-1868) and were in accordance with the ARRIVE criteria for reporting animal studies [18,19]. Female albino Wistar rats were born and maintained in the animal facility of the INCMNSZ, which is accredited by the Association for Assessment and Accreditation of Laboratory Animal Care International (AAALAC) and adheres to its requirements. Rats were maintained under a 12 h light–dark cycle (lights on from 07:00 to 19:00 h) at a constant temperature (22–23 °C) with food and water given ad libitum.

### 2.2. Experimental Design

To produce Founder Generation (F0) mothers, 120-day-old female Wistar rats were randomly mated with proven fertile non-littermates. At birth (day 0), F0 litters were adjusted to ten pups, including at least four females [2]. At weaning (21-day-old), F0 females were randomly assigned to one of the two experimental groups: the control (F0C) or maternal obesity (F0MO) group to be fed either a standard laboratory chow diet or a high-fat diet (HFD) (Figure 1). The C diet consisted of standard laboratory chow (Zeigler Rodent RQ22-5, Gardners, PA, USA) containing 22.0% protein, 5.0% soy oil fat, 31.0% polysaccharide, 31.0% simple sugars, 4.0% fiber, 6.0% minerals and 1.0% vitamins (*w/w*) (physiological fuel 3.4 kcal/g). The HFD was produced in the specialized dietary facility of the INCMNSZ with 23.5% protein, 20.0% lard, 5.0% soy oil fat, 20.2% polysaccharide, 20.2% simple sugars, 5.0% fiber, 5.0% mineral mix, 1.0% vitamin mix (*w/w*), physiological fuel 4.8 kcal/g.

Based on the fertility rate from previous studies in this model [2] at 120 days, 10 female rats from the F0C group and 20 from the F0MO group were mated overnight (up to 5 days) with non-experimental males to generate offspring (F1). Daily vaginal smears were obtained, and the day in which a sperm plug was found was designated as day 0 of conception. F0C and F0MO continued to be fed with C or HFD, respectively, throughout pregnancy and lactation (Figure 1). The fertility rate for F0C was 80% and, for F0MO, 40%.

To achieve F1 homogeneity, on postnatal day (PND) 2, all F1 litters studied were adjusted to 10 pups, with equal numbers of males and females whenever possible (F1C and F1MO). Litters with less than 9 or more than 14 pups were excluded from the study. F1 litters were weaned at PND 21, housed 2–3 pups per cage, and fed a control diet until the end of the experiment (PND 110).

### 2.3. F1: Food Intake

From PND 95 to 110, offspring food intake was recorded for 15 consecutive days. A maximum of three rats from the same sex and experimental group (F1C or F1MO) were housed per cage. The daily amount of food provided was weighed, as was the amount that remained after 24 h. The amount of food consumed daily was averaged to provide a value per rat. Food intake was reported as the 15-day average.

### 2.4. F1: Blood and Tissue Collection

At PND 110, between 12:00 and 14:00 h, after 6 h of fasting, one male and female F1 per litter (*n* = 8) were euthanized by exsanguination through aortic puncture under general anesthesia with isoflurane by the same person under identical conditions (Figure 1). Blood was collected, and mediastinal, retroperitoneal, omental, mesenteric, parametrial and periovarian or epidydimal fat depots were excised and weighed to determine the total visceral adipose tissue. One sample of the retroperitoneal fat pad was frozen for molecular analysis, whereas another sample from the same pad and region was fixed for histological analysis.

### 2.5. F1: Biochemical Parameters

Blood samples were centrifuged for 15 min at 4 °C at 2880 RCF, and the serum was stored at −70 °C until biochemical and hormonal analysis. Glucose (GLUH, # B24985 Beckman Coulter, Inc., Brea, CA, USA) and triglyceride (TG, #445850 Beckman Coulter, Inc., Brea, CA, USA) concentrations were determined enzymatically with an automatic analyzer (Synchron CX Beckman Coulter, Brea, CA, USA). Insulin serum concentrations were determined by radioimmunoassay (Millipore Radioimmunoassay kit, Burlington, MA, USA). A homeostatic model assessment for insulin resistance (HOMA-IR) was calculated using HOMA-IR= [glucose (mmol/L) × insulin (μU/mL)]/22.5 [20,21].

### 2.6. F1: Adipose Tissue Histology

Retroperitoneal AT samples were fixed using 10% paraformaldehyde in PBS 0.05M, which were dehydrated and paraffin-embedded. A total of 5 μm thick tissue sections were mounted on poly-L-lysine-coated slides. The slides were deparaffinated, rehydrated, and then stained with hematoxylin and eosin [22]. Histology slides were examined under a light microscope (Olympus BX51, Melville, NY, USA) at 20×magnification. AS was measured manually by delimiting the adipocyte cross-sectional area in digital images using AxioVisio LE software real 4.8 version (Zeiss^®^ copyright 2006–2010 Stuttgart, Germany) in at least 180 cells per group corresponding to an average of 25 cells per animal. All histological measurements were performed by an observer blinded to the nature of the tissue source [6].

### 2.7. F1: Adipocyte Size Distribution

The cross-sectional area of the adipocytes was measured in μm^2^. Histograms of the relative frequency of 500 μm^2^-area intervals were overlaid with their corresponding representative gamma distribution functions (GDFs). The AS distribution analysis, based on GDF modeling, was performed as previously reported [13]. Briefly, GDFs were modeled using parameter estimators of shape (α) and scale (β), which were calculated as: α = (Mean/Standard Deviation)^2^ and β = (Standard Deviation^2^/Mean); and plotted using the GAMMA.DIST function (Microsoft^®^ Excel^®^ for Microsoft 365 MSO Version 2304 Build 16.0.16327.20200 64-bit). The 10th and 90th percentiles of the GDFs of C groups were used to determine the cut-off points for small and large adipocytes, respectively. For each rat, the proportions of small and large AS were calculated from their GDFs as the % of cumulative probability below and above small and large AS cut-off points.

### 2.8. F1: Retroperitoneal Adipose Tissue RNA Extraction and Complementary cDNA Synthesis

A total of 0.1 g frozen retroperitoneal AT samples were homogenized with 0.8 mL of a Tri-Reagent (Zymo, TRI Reagent^®^ #R2050-1-200, Irvine, CA, USA) using a rotor/stator homogenizer (Dremel, Tissue Tearor #985-370, Bartlesville, OK, USA) at 2/3 of the maximum speed setting for 5–10 s in an ice bath. Homogenates were centrifuged at 12,000× *g* RCF for 10 min at 4 °C; the surface’s fatty layer was removed, and the supernatant was transferred into an RNAse-free tube and purified with an RNA Column Purification Kit (Zymo, Direct-zol RNA Mini-Prep #R2053, Irvine, CA, USA) in accordance with the manufacturer’s specifications.

The purity and concentration of the total RNA were determined using a microvolume UV/Visible Spectrophotometer (Thermo Scientific, NanoDrop 2000 #ND-2000, Waltham, MA, USA). Electrophoresis chips (Agilent Technologies, RNA 6000 Nano Kit #5067-1511, Santa Clara, CA, USA) were used to determine the RNA integrity number (RIN) on a Bioanalyzer System (Agilent Technologies, Agilent 2100 #G2940AA, Santa Clara, CA, USA). Samples with: A260/A280 and A260/A230 ratios ≥ 1.8 [23] and no evidence of ribosomal peak degradation with a RIN ≥ 6.0 [24,25,26] were used for expression assays.

The total-RNA samples of each rat were processed using a cDNA Conversion Kit (Qiagen RT^2^ First Strand Kit #330401, Germantown, MD, USA) in accordance with the manufacturer’s specifications. Briefly, for each sample the genomic DNA elimination mix containing 500 ng of the total RNA was incubated at 42 °C for 5 min, followed by 15 min of reverse transcription at 42 °C, and was stopped for 5 min at 95 °C.

### 2.9. F1: Adipose Tissue Expression Profile

Eighty-four adipogenesis-related mRNA transcripts (Appendix A) were analyzed using the SYBR Green qPCR Mastermix (Qiagen, RT^2^ SYBR Green ROX #330522, Germantown, MD, USA) and a qPCR array (Qiagen, RT^2^ Adipogenesis Profiler PARN-049Z #330231, Germantown, MD, USA) in a multi well-plate RT-PCR system (Applied Biosystems, Step One Plus #4376600, Waltham, MA, USA) in accordance with the manufacturer’s specifications. In brief, plated mRNA transcripts were incubated at 95 °C for 10 min to activate DNA Taq Polymerase, followed by 40 cycles of 15 s at 95 °C and 1 min at 60 °C for the collection of fluorescence data.

Gene expression was analyzed by the 2^−∆∆Ct^ method and Student’s *t*-test in the Qiagen RT^2^ Profiler PCR Analysis Web Portal https://dataanalysis2.qiagen.com/pcr, accessed on 9 August 2020. RT-qPCR arrays included 5 candidate housekeeping genes (HKG), and *Hprt1* was selected among *Actb*, *B2m*, *Ldha*, and *Rplp1* due to its expression stability. Gene expression with a fold-change of (FC) ≥ 1.4 and a statistical significance of *p* < 0.05 were considered as differentially expressed genes (DEGs) [4,27,28]. Common and exclusive DEG in different groups were identified using the Venny online software https://bioinfogp.cnb.csic.es/tools/venny/index.html, accessed on 22 December 2020 [29].

#### F1: Differentially Expressed Genes Interaction

Potential functional interactions of DEGs at the protein level were predicted in the F1C female and F1MO male. DEGs Refseq accession numbers were uploaded to the Search Tool for the Retrieval of Interacting Genes/Proteins (STRING) database at https://string-db.org, accessed on 12 January 2021 [30]. DEG interaction networks were derived from validated interaction sources, including text mining, experiments, databases, co-occurrence, and co-expression, applying an interaction score > 0.4. The KEGG identification pathway was set in the criteria to identify the Predicted Protein Interaction (PPI) in relation to adipose tissue development and function.

### 2.10. Statistical Analysis

Physiological and biochemical parameters were expressed as the mean ± standard error of the mean (SEM) and were assessed by a one-tailed *t*-test, alongside a median AS and small and large adipocyte proportions by the Mann–Whitney U-test. These analyses were performed using Sigma Plot 11.0 software. Cumulative AS were compared through a Kolmogorov–Smirnov test using GraphPad Prism 7.04 software. Gene expressions were compared by a *t*-test using the Qiagen RT^2^ Profiler PCR Analysis Web Portal. Statistically significant differences were defined as *p* < 0.05. F1C male and female body weight and adipose tissue mass, which are basic and well-characterized biologically meaningful endpoints, were analyzed by a one-sided *t*-test and were different (*p* < 0.05); therefore, the sexes were analyzed separately [31].

## 3. Results

### 3.1. F1: Food Intake and Adiposity

F1C males had increased body mass, caloric intake, and total fat compared to F1C females (Figure 2D–F). Body weight (Figure 2A,D) and food intake (Figure 2B,E) were similar in both F1C and F1MO males and females. However, adipose tissue mass was higher in both male and female F1MOs when compared to F1C (Figure 2C,F).

#### F1: Male and Female Fat Depots

F1C males had increased retroperitoneal, mesenteric, omental and mediastinal fat compared to F1C females. Omental and mediastinal (*p* = 0.056; F1C vs. F1MO males) fat were similar in both F1C and F1MO males and females; epididymal fat was similar between male F1C and F1MO; retroperitoneal, parametrial and mesenteric fat depots were similar in female F1C and F1MO. F1MO males had increased retroperitoneal and mesenteric fat depots compared to F1C males, and F1MO females had increased periovarian fat compared to F1C females (Table 1).

### 3.2. F1: Biochemical Parameters

F1C females had higher serum glucose concentrations than F1C males (Figure 3A,F). Serum glucose concentrations were higher only in F1MO males vs. F1C males (Figure 3A,F). However, insulin (Figure 3B,G), HOMA-IR (Figure 3C,H), triglyceride (Figure 3D,I) and leptin (Figure 3E,J) serum concentrations were increased in both male and female F1MO in comparison to the F1C male and female, respectively.

### 3.3. F1: Adipocyte Size Distribution

Cumulative AS showed a shift to smaller size F1C females compared to F1C males. In contrast, there was a shift to a greater size in F1MO females and males compared to respective F1C. For both males and females, the median AS was significantly greater in F1MO than in F1C (Figure 4A,B). In all groups, the observed AS distribution fit to gamma distribution exhibited a peak located at the left of the center and a more gradual tapering to the right with more dispersion in F1MO than in F1C males and females (Figure 4C,D). Adipocyte size cut-off points were established at 1022 μm^2^ for small and 5830 μm^2^ for large adipocytes in males, and 813 μm^2^ for small and 4736 μm^2^ for large adipocytes in females (Figure 4C).

### 3.4. F1: Adipogenic Expression Profile

The adipogenic expression profile (Appendix A) showed a total of 65 DEGs in F1C females vs. F1C males (Figure 5A), 58 in F1MO males vs. F1C males (Figure 5B) and 1 in F1MO females vs. F1C females (Figure 5C). All DEGs were downregulated in the F1C female vs. F1C male and F1MO male vs. F1 male, while the 10 most significant were: *Retn*, *Slc2a4*, *Fasn*, *Insr*, *Tsc22d3*, *Nr1h3*, *Lipe*, *Cebpa*, *Ncor1*, and *Mapk14*. Only *Egr2* expression was downregulated in the F1MO female vs. F1C females (Table 2).

The union and intersection analysis of DEG sets showed that 56 DEGs were regulated by sex as well as maternal diet, 9 DEGs (*Bmp4*, *Fos*, *Runx1t1*, *Klf4*, *Ddit3*, *Bmp7*, *Sfrp1*, *Pparg*, and *Adipoq*) were regulated only by sex, 2 DEGs (*Gata3* and *Irs1*) were regulated only by maternal diet within males, and 1 DEG (*Egr2*) was regulated only by the maternal diet within females (Figure 5D).

#### Differentially Expressed Genes Interaction

According to STRING v11.0 PPI analysis, significant connectivity among DEGs was found as F1C females showed 62 genes with 345 interactions, and F1MO males had 56 genes with 268 interactions (PPI enrichment *p* < 1.0 × 10^−16^). Similar gene sets in F1C females vs. F1C males; F1MO males vs. F1C males were involved in the Wnt pathway, 11 and 13 DEGs, respectively, and in insulin signaling pathways, 10 and 13 DEG, respectively (Figure 6). However, F1C females showed 13 DEGs to be involved in the AMPK signaling pathway with a false discovery rate (FDR) ≤ 8.04 × 10^−6^ (Figure 6A), while F1MO males showed 14 DEGs involved in PI3K-Akt signaling with an FDR ≤ 1.56 × 10^−6^ (Figure 6B).

## 4. Discussion

There is now compelling evidence that environmental factors, such as inadequate maternal nutrition during pregnancy and/or in early neonatal life, may increase offspring’s susceptibility to developing type 2 diabetes, hypertension, and obesity [32,33,34,35]. We previously reported that obese mothers show metabolic dysfunction at 19 days of gestation [36], at the end of lactation [2] and that milk production and the % of water and carbohydrate content were lower in obese mothers, whereas milk leptin, total fat, arachidonic acid and monosaturated fatty acids were higher, with no changes in protein content [37] explaining the adverse metabolic outcomes of offspring. However, in many studies, sex differences in offspring are often overlooked as a significant variable in disease development. In the present study, we observed sexual dimorphism in both F1C males and females and between F1MO and F1C. These findings are consistent with our previous studies, in which the male offspring of obese mothers but not females had greater fat mass, leptin, and triglyceride concentrations at PND 36 [2]. In addition, we reported that MO programs insulin, glucose, and lipid signaling in the liver of young adult offspring in a sex-dependent manner, leading to liver dysfunction and insulin resistance [3,4] and that the male and female offspring of obese mothers have different aging metabolic trajectories [3].

Body fat distribution differs by sex and is a significant indicator of health; the accumulation of retroperitoneal adipose tissue, for example, raises the risk of developing cardiometabolic diseases. Sex variations are due to the interaction of sex chromosomes, genetic diversity, sex hormones, and the environment. Therefore, a better understanding of the molecular mechanisms driving sexual dimorphism could contribute to the development and implementation of interventions to prevent metabolic and cardiovascular disease in offspring born to obese mothers [38]. Several studies have proposed that male and female offspring have different susceptibilities to metabolic alterations in relation to insulin resistance in response to maternal HFD consumption [3,5], although the reasons for this difference are unclear. After sexual maturity, the expansion of retroperitoneal adipose tissue plays a crucial role in the development of insulin resistance [16,39]. Therefore, the observed metabolic differences between male and female F1MO young adults suggest that the expansion of retroperitoneal adipose tissue is one of the mechanisms by which MO programs the offspring’s metabolic phenotype in a sex-dependent manner.

Increased insulin concentrations, HOMA-IR, and adiposity were observed in male and female F1MO, with augmented serum glucose in males but not in females. Previous reports indicated that hyperinsulinemia and insulin resistance developed before the onset of obesity in F1MO [40]. However, the failure to observe elevated glucose concentrations in F1MO females despite having increased visceral adiposity may be related to the increased expandability capacity of the female retroperitoneal adipose tissue due to an increase in the number of adipocytes [41,42]. In this regard, adipose tissue hyperplasia has been significantly associated with better glucose and insulin concentrations than adipose hypertrophy [43]. Therefore, the increased proportion of small adipocytes in females vs. males might contribute to the reduced effect of insulin resistance in young adult females [44]. Several studies have shown that, in contrast to males, females who gain more relative body weight or adiposity in response to HFD do not develop hyperglycemia and are more glucose tolerant [45,46,47]. The association of higher adipose tissue expandability capacity, due to increased adipocyte precursor proliferation, with improved glucose homeostasis is not well understood, though it likely involves several mechanisms, such as protection from adipose tissue hypertrophy and reduced ectopic lipid accumulation [48].

In accordance with a previous systematic review and meta-analysis [49], triglyceride serum concentrations increased similarly in F1MO males (32%) and females (36%) when compared to the respective F1C groups. As triglyceride serum concentration reflects the lipid rate of entry and removal into the blood, hypertriglyceridemia may be considered the central pathophysiologic feature of the abnormal lipid profile. Although increased serum triglyceride is frequently associated with dietary intake [50], in the present study, all F1MO were weaned to the same C diet; therefore, the main factors leading to increased serum triglyceride in F1MO may be due to perturbation in lipolysis and lipogenesis balance in the liver and adipose tissue, which determine the lipid uptake and its release to the blood [4].

Maternal diet is important to offspring health, but the postnatal environment also influences the phenotype. The mice offspring of obese mothers (first-hit) were fed a high-fat diet (second-hit) after weaning and exhibited triglyceride profile changes between adipose depots and sexes [51]. We have published that the postnatal consumption of a high-fat diet by the offspring of obese mothers exacerbates body fat accumulation as well as a decrease in small and an increase in large adipocytes in retroperitoneal adipose tissue, resulting in larger metabolic abnormalities [14].

Previous studies using the same experimental model (young adult offspring) showed marked characteristics of fatty liver in males but not in females F1MO [4]. The increased liver weight, total fat, and the triglyceride content seen in F1MO males was attributed to a decrease in hepatic lipid metabolism capacity. In contrast, the slight rise in liver triglyceride concentration found in F1MO females was attributed to enhanced hepatic de novo lipogenesis, as shown by liver transcriptome profiles [4]. Our previous results [4] suggest that a reduced lipolysis rate in the liver could significantly contribute to increased triglyceride concentrations in male and female F1MO, which cannot be counterbalanced by liver lipid uptake and lipolysis and might be worsened by increased hepatic lipogenesis and lipid release in F1MO females. Impaired lipid mobilization from adipocytes is associated with higher lipogenesis in the liver as a result of insulin’s failure to regulate lipolysis in the adipose tissue [52].

Histologically, we found retroperitoneal adipose tissue hypertrophy in both male and female F1MO, which was consistent with prior studies [13,53,54,55]. It has been shown that adipose tissue hypertrophy and insulin resistance indicates the presence of lipid-overloaded adipocytes in the offspring of obese mothers [56]. Although F1MO males and females exhibited a greater median AS, these changes themselves were not capable of explaining the different metabolic effects observed. Due to the heterogeneity of AS distribution among the groups, cumulative frequencies differed more than the median AS; consequently, a better approach to understanding the AS would be to analyze the relative frequencies that are characterized by heavy and light tails. Gamma distribution modeling provides a statistical method to characterize AS distribution. In this work, the retroperitoneal adipose tissue of MO males exhibited a greater proportion of large adipocytes than MO females, which could be related to a sex-specific offspring response to the maternal nutritional state.

The AS fit to a gamma distribution model in all groups, agreeing with previous works [13,57], in which the analysis of AS distributions revealed that they were more asymmetric in F1C groups and less asymmetric and with more spread in F1MO males and females compared to F1C. The AS distribution asymmetry was characterized by a short and heavy tail on the left, and a long and thin tail on the right, which could be described as having a short-range size for small adipocytes and a long-range size for large adipocytes in all groups. The small size was associated with newly generated adipocytes, and large sizes were related to adipocyte lipid storage capacity. Small and large adipocyte proportions are linked to adipocyte precursor proliferation and adipocyte late differentiation, respectively. Therefore, adipocyte extreme sizes located at AS distribution tails are important parameters for evaluating the mechanisms of expansion in the adipose tissue at the cellular level [13]. Currently, it is recognized that the proportions of different adipocyte size range, namely small, large, and very large, are indications of the metabolic state [58]. Large adipocytes are linked to energy storage, whereas small adipocytes are connected to lower insulin resistance [59,60].

In the present study, the histological analysis of retroperitoneal adipose tissue samples from F1 rats revealed an AS cumulative distribution shift toward smaller sizes, leading to a lower median AS in F1C females compared to males. Consequently, the adipocyte cut-off for a small size in males increased by 26% compared to females, and the cut-off value for large adipocytes was 23% greater in males than in females, indicating that females under physiological conditions had a higher proportion of small adipocytes in a relative short size range and a lower proportion of large adipocytes in retroperitoneal adipose tissue than males, suggesting a more proliferative adipose tissue capacity in females than males. It is well-accepted that estradiol stimulates the proliferation of preadipocytes in both males and females; however, female preadipocytes are more responsive to estradiol and proliferate more rapidly than male preadipocytes [61]. According to experimental models using 3T3-L1 cells, it has been shown that estradiol facilitates adipocyte proliferation and differentiation [62]. Therefore, the increased abundance of small adipocytes in F1C females to F1C males may be due to female estradiol levels. This is in agreement with human studies, in which the preadipocyte proliferation rate is stimulated more by estradiol in retroperitoneal adipose tissue in women than in men [63]. Supporting these findings, ovarian hormone withdrawal models with ovariectomized (OVX) rats exhibited a higher body weight and visceral adiposity, while OVX rats with estradiol replacement exhibited a substantial body weight decrease with a modest tendency to improve insulin sensitivity [41,64]. In another study, OVX rats with and without estradiol replacement showed similar AS in retroperitoneal fat depots [64], suggesting that the observed differences between F1C males and females in AS distributions in the present study could be attributed to other sex-related factors than estradiol levels. In this regard, the observed decrease in the AS cut-off for small adipocytes in females respective to males could be related to the increased differentiation of preadipocytes, confirming the female’s higher adipogenic capacity compared to males.

In the present study, the adipogenic expression profile of F1C females vs. F1C males reflected a greater expansion capacity in female than male retroperitoneal adipose tissue, and an increased amount of adipocyte precursors and newly differentiated adipocytes contributed, at least partially, to a lower adipocyte lipid accumulation and led to a shift in AS distribution to a smaller size [42]. Wnt signaling plays a central role in adipose tissue development, which is required for lipogenic gene expression during terminal adipocyte differentiation via β-catenin-signaling and for regulating de novo lipogenesis by *Fasn* expression [65,66,67]. The downregulation of genes associated with *Wnt* signaling observed in F1C females vs. F1C males was in line with prior rodent studies that identified the sexually dimorphic expression of genes related to adipose tissue morphogenesis [68]: therefore, the lower *Fasn* expression observed in C females vs. C males, was related to *Wnt* signaling disruption.

AMPK signaling is a fuel-sensing enzyme and is a major regulator of cell cycle and energy homeostasis [69]. AMPK activation in adipose tissue was linked to the inhibition of white preadipocyte differentiation [70]. In the current study, AMPK signaling genes were downregulated in F1C females vs. F1C males; this downregulation in females suggested a more active preadipocyte differentiation than in males. Previous studies showed that *Retn* expression was higher in males than females, induced during adipogenesis and elevated in obesity [71], which is in line with our findings in which *Retn* gene expression was lower in F1C females compared to F1C males, suggesting a higher rate of preadipocyte differentiation in females. The solute carrier family 2 (facilitated glucose transporter), member 4 (*Slc2a4*), previously known as *Glut4*, which is encoded by the *Slc2a4* gene, is predominantly expressed in adipose tissue and muscle. Nevertheless, *Slc2a4* expression during adipocyte development is extremely low, particularly in undifferentiated precursor cells [72]. The observed reduced expression of *Slc2a4* in F1C females compared to F1C males could be associated with an increased content of adipocyte precursors in adipose tissue samples rather than an impairment of insulin signaling, given that there was no evidence of insulin resistance in terms of HOMA-IR.

Clearly, under physiological conditions, there are sexual differences in the mechanisms that control the expansion of adipose tissue. Therefore, suboptimal conditions during pregnancy and lactation, such as maternal obesity, lead to sex-specific changes in the expression of adipogenesis-related genes in offspring [6]. Lecoutre et al. showed that maternal obesity in the rat programs increased adiposity in a sex-specific manner and that perirenal adipose tissue was more sensitive than gonadal adipose tissue to maternal obesity programming, as evidenced by adipocyte hypertrophy and a decrease in adipogenic genes (*Pparγ*) in male offspring [6]. However, the molecular mechanisms that explain the insulin resistance of male progeny of obese mothers have not yet been fully understood. In the present study, the adipogenic profile identified genes associated with insulin resistance that were substantially related to one another (Figure 6). Our research supports the fact that sex-specific insulin resistance in response to maternal obesity originates predominantly in adipose tissue [12]; however, additional evidence is required to confirm this hypothesis.

Although adipogenesis regulation is strongly linked to *Pparγ* expression, this process is mediated by the recruitment of its co-activators and co-repressors, which are considered the transcriptional switches for adipocyte differentiation and lipogenesis programmed in the offspring of obese mothers [73]. MO leads to epigenetic modifications in the main adipogenesis regulator genes in offspring. The adipose tissue of male rat offspring from obese mothers exhibited hypermethylation and histone modification of the *Pparγ2* promoter, which led to *Pparγ2* downregulation, supporting the cellular memory concept in which adipocyte precursors retained epigenetic marks, predisposing offspring to obesity [74,75,76]. Indeed, the DNA methylation of the *Zfp42* promoter has been reported to decrease in the adipose tissue of male offspring of obese mothers [77]. Microarray expression analysis demonstrated that the increased expression of insulin signaling cascade genes in female adipose tissue contributed to reduced insulin resistance and diabetes risk compared to males [78].

Our results showed that in MO males compared to C males, the retroperitoneal AT downregulation of *Wnt* signaling was accompanied by pronounced hypertrophy and indicators of insulin resistance. Several studies employing rodent obesity models have supported the notion that *Wnt* signaling is suppressed during the hypertrophic stages [12,70,71,72]. A mice study reported that after several weeks of consuming a high-fat diet, the adipocyte precursors remained quiescent, and the suppression of adipocyte precursor proliferation was associated with *Wnt* signaling inhibition and adipose tissue hypertrophy [79]. Other studies have suggested that the activation of the PDGFRα pathway in adipose tissue suppresses *Wnt/β-catenin* signaling during the final stages of adipogenesis, which has been linked to an early onset of insulin resistance [80,81]. Indeed, a reduction in the Wnt signaling-related expression was observed in young healthy individuals with certain low insulin sensitivity [82]. Alterations in *Wnt* signaling have also been reported to cause insulin resistance in cultured preadipocytes, indicating a crosstalk between *Wnt* and insulin signaling [83].

The PI3K/AKT pathway is crucial for insulin action in adipose tissue, and its proper signaling is required for insulin sensitivity [84]. In fact, several anti-cancer PI3K inhibitors induce hyperglycemia and insulin resistance [85]. In accordance with the adipogenic expression profile, several PI3K/AKT signaling-related genes were downregulated in F1MO males compared to F1C males. The observed insulin resistance in F1MO males may be mediated by an impairment of PI3K/AKT signaling.

The *Egr2,* formerly known as *Krox20,* encodes the Early Growth Response 2: a zinc-finger transcription factor that is necessary for adipogenesis [86]. The present work showed *Egr2* downregulation with a reduction in the small adipocyte proportion in F1MO females vs. F1C females, supporting the association between *Egr2* suppression and lower adipocyte precursor differentiation. However, the reduction in early adipogenesis was compensated by an increased lipid accumulation in differentiated adipocytes, resulting in a greater proportion of large adipocytes in F1MO females compared to F1C females.

In young adults, F1MO females exhibited fewer adverse metabolic outcomes than F1MO males, as explained by the increased adipogenesis. However, metabolic alterations worsened with age in the male and female F1 of obese mothers [3] due to a subsequent depletion in the adipocyte progenitor pool [11]. Other authors have reported that adipose tissue gene expression exhibited sexual dimorphism regardless of diet, fat depot, and function [78]. Growing data suggest that sex-specific differences in the functionality of each kind of adipose tissue were reliant not only on the amount of a particular fat depot but also on their differential gene expression patterns [87,88]. Differences in sex hormone concentrations, such as estradiol, and the amount and sensitivity of its receptors expressed in adipose tissue led to the sex differences of master adipogenesis regulator gene expression, such as *Pparγ* [61]. However, the sex chromosome complement has been proposed as another determinant of sex morphologic and physiologic differences in WAT. The proposed mechanism driving these differences involves the inactivation and imprinting of X chromosomal genes [89] and miRNA regulation of adipogenesis [90], but further studies providing additional insights on AT sexual dimorphism are needed.

## 5. Conclusions

Maternal obesity leads to sex-specific changes in AS distribution and adipogenic profile in offspring, and these differences can help explain the intergenerational transmission of metabolic alterations. Reducing the transmission of obesity and its metabolic comorbidities, such as insulin resistance and dyslipidemia, is a significant challenge for the scientific community. Consequently, a greater understanding of the mechanisms involved in the programming and regulation of adipocyte gene expression is needed. In the present study, adipogenic profile analysis allowed us not only to identify molecular differences in the programming mechanism of adipose tissue expansion, such as the decreased expression of pro-adipogenic genes in F1MO males and lipid mobilization-related genes in F1MO females (Figure 7) but also that the mechanisms implicated in the metabolic alterations of F1MO were sex-specific. In F1MO males, this was mainly due to decreased insulin signaling in retroperitoneal adipose tissue. Therefore, interventions to counteract the adverse metabolic effects observed in F1MO should be designed in accordance with the programmed metabolic risk underpinning the programmed adipose tissue mechanism of expansion in response to maternal obesity.

## Figures and Tables

**Figure 1 nutrients-15-02245-f001:**
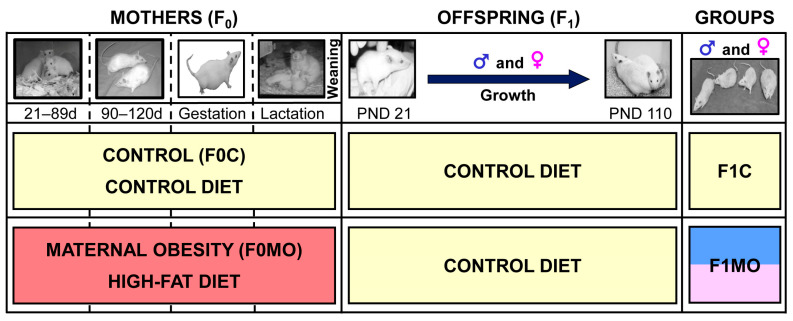
Experimental design. Timeline for the study of maternal (F0) and offspring (F1) experimental groups.

**Figure 2 nutrients-15-02245-f002:**
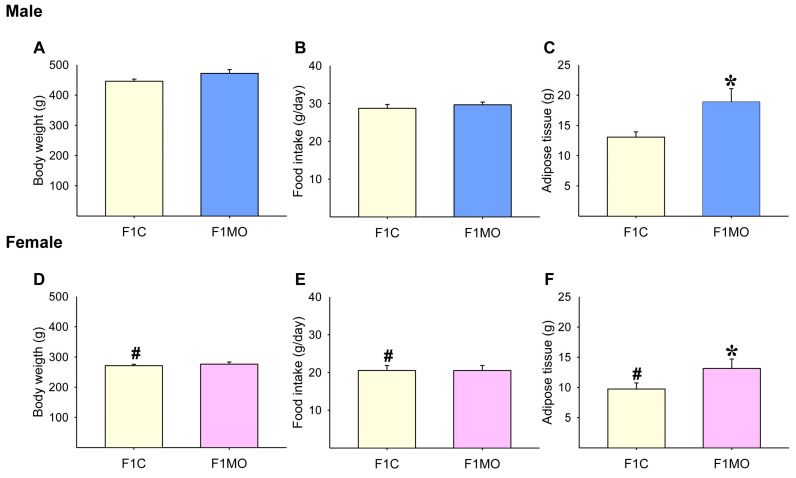
Adult offspring morphometric parameters. Male and female: body weight (**A**,**D**), food intake (**B**,**E**), and adipose tissue mass (**C**,**F**). Data are mean ± SEM, *n* = 8 rats per group, # *p* < 0.05 vs. F1C male, * *p* < 0.05 vs. F1C.

**Figure 3 nutrients-15-02245-f003:**
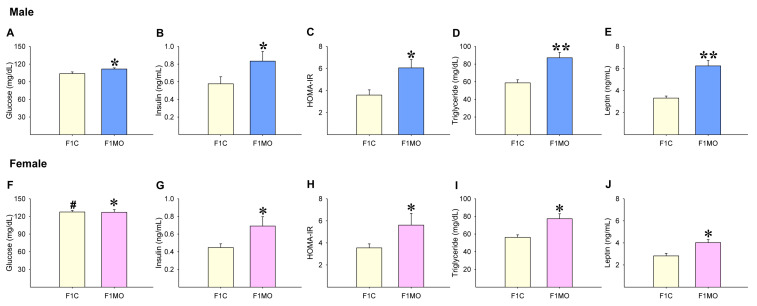
Offspring serum metabolic parameters. Male and female: glucose (**A**,**F**), insulin (**B**,**G**), HOMA-IR (**C**,**H**), triglyceride (**D**,**I**) and leptin (**E**,**J**). Mean ± SEM, *n* = 8 rats per group, # *p* < 0.05 vs. F1C male, * *p* < 0.05, ** *p* < 0.01 vs. F1C.

**Figure 4 nutrients-15-02245-f004:**
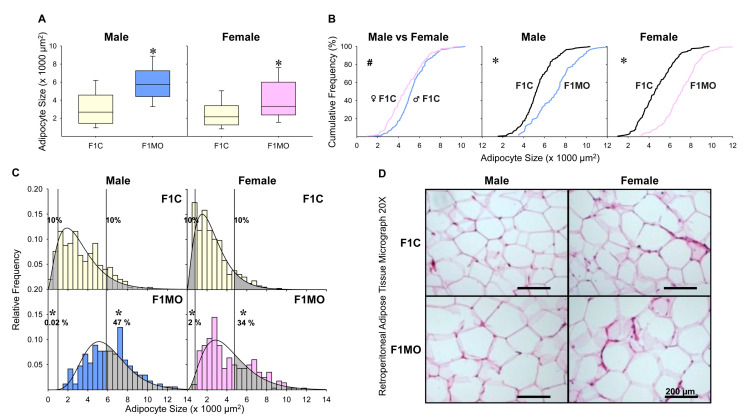
Offspring adipocyte size distribution. Male and female median adipocyte size, with their interquartile range, 10th and 90th percentile (**A**). Adipocyte size cumulative distributions (**B**), relative frequency histograms with their representative gamma distribution functions with small and large adipocyte proportions shown as shaded regions (**C**). Representative H&E-stained photomicrographs of retroperitoneal adipose tissue at 20× magnification (**D**). The Mann–Whitney test was used to compare median adipocyte size and proportions of small and large adipocytes, and the two-sample Kolmogorov–Smirnov test was used to compare the cumulative distributions. Small and large adipocyte size cut-off points were defined as the 10th and 90th percentile of F1C groups. *n* = 8 per group. # *p* < 0.05 vs. F1C male; * *p* < 0.05 vs. F1C.

**Figure 5 nutrients-15-02245-f005:**
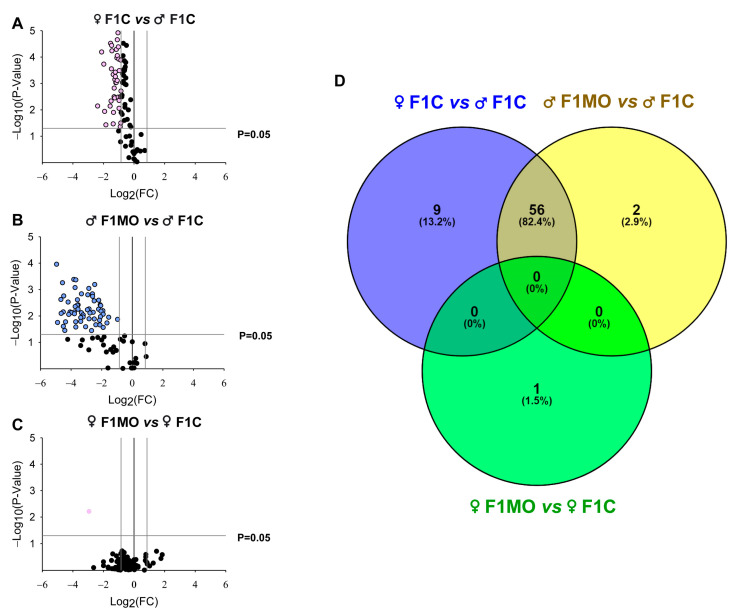
Offspring adipogenic expression profile and Venn diagram of differentially expressed genes (DEGs). ‘Volcano Plots’ showing the gene expression in (**A**) F1C female, (**B**) F1MO male and (**C**) F1MO female. Downregulated (pink and blue dots) and non-differentially expressed (black dots) genes are shown. Fold-change (FC = 1.5) and statistical significance (*p* = 0.05) cut-off values (gray lines) are indicated. Expression profile analyzed in RT-qPCR arrays of 84 genes related to adipogenesis (RT^2^ Profiler PCR array PARN-Z49, Qiagen). (**D**) Overlapping of DEGs sets in F1C female (blue), F1MO males (yellow) and F1MO females (green) are shown.

**Figure 6 nutrients-15-02245-f006:**
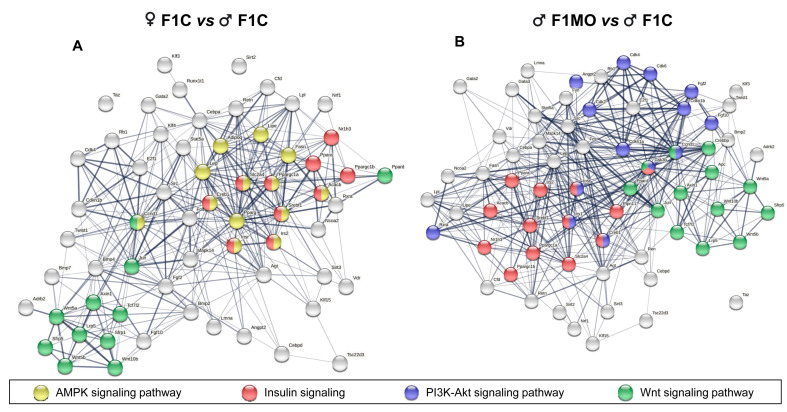
Functional interaction of differentially expressed genes (DEGs). Known and predicted protein–protein interactions (PPI) of downregulated DEGs in (**A**). F1C females and (**B**). F1MO males. Interactions (lines) between DEGs (nodes) and the association strength (line thickness) are shown. KEGG pathways related to adipose tissue development and function are indicated (highlighted nodes). PPI network analysis performed in STRING v11.0.

**Figure 7 nutrients-15-02245-f007:**
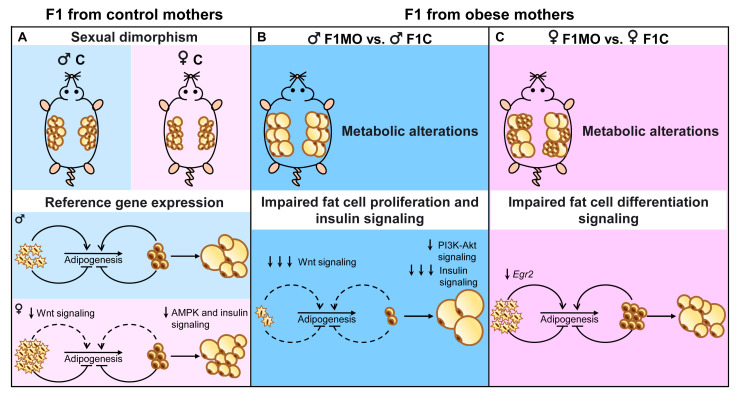
Adipogenic sexual dimorphism in F1C and F1MO. Maternal obesity programs in a sex-specific manner adipocyte size and gene expression, suggesting a down-regulation of adipogenesis in males and down-regulation of lipid mobilization in females.

**Table 1 nutrients-15-02245-t001:** Fat depots in male and female F1C and F1MO at 110 PND.

Fat Depot (g)	F1C	F1MO
	Male	
Mediastinal	0.2 ± 0.02	0.7 ± 0.3
Retroperitoneal	5.1 ± 0.5	8.2 ± 1.4 *
Omental	0.6 ± 0.1	0.7 ± 0.1
Mesenteric	2.5 ± 0.3	3.4 ± 0.4 *
Epidydimal	5.4 ± 0.3	6 ± 0.6
	Female	
Mediastinal	0.1 ± 0.02 #	0.3 ± 0.2
Retroperitoneal	2.7 ± 0.2 #	3.1 ± 0.4
Omental	0.4 ± 0.1 #	0.6 ± 0.1
Mesenteric	1.6 ± 0.2 #	2.1 ± 0.3
Parametrial	3.4 ± 0.5	4.3 ± 0.6
Periovarian	1.5 ± 0.2	2.4 ± 0.4 *

Data are mean ± SEM, *n* = 8 rats per group, # *p* < 0.05 vs. F1C male, * *p* < 0.05 vs. F1C.

**Table 2 nutrients-15-02245-t002:** Top 10 differentially expressed genes (DEG) related to adipogenesis in F1C females vs. F1C males and F1MO males vs. F1C males. The 10 most significant (by *p*-value) and common DEGs in F1C females and F1MO males are shown. F1MO females vs. F1C females only exhibited one single DEG.

		F1C		F1MO vs. F1C	
**Gene**	Description	Fold Regulation	*p*-Value (*t*-Test)	Fold Regulation	*p*-Value (*t*-Test)
		**Female vs. Male**		**Male**	
*Retn*	Resistin	−65.4	0.000001	−7.2	0.0009
*Slc2a4*	Solute carrier family 2, member 4	−44.9	0.000001	−30.9	0.0001
*Fasn*	Fatty acid synthase	−29.1	0.00004	−14.2	0.0004
*Insr*	Insulin receptor	−27.9	0.00005	−25.4	0.002
*Tsc22d3*	TSC22 domain family, member 3	−14.3	0.00005	−11.6	0.004
*Nr1h3*	Nuclear receptor 1H3	−12.4	0.00008	−13.4	0.0008
*Lipe*	Lipase, hormone sensitive	−11.2	0.00001	−12.9	0.0004
*Cebpa*	CCAAT/enhancer binding protein α	−10.6	0.00004	−7.5	0.002
*Ncor1*	Nuclear receptor co-repressor 1	−5.2	0.00009	−6.0	0.002
*Mapk14*	Mitogen activated protein kinase 14	−5.0	0.00003	−5.5	0.0006
				Female	
*Egr2*	Early growth response 2			−7.6	0.006

## Data Availability

The data supporting the research for this study are available within the manuscript.

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
