# Peer review of "Programming Mechanism of Adipose Tissue Expansion in the Rat Offspring of Obese Mothers Occurs in a Sex-Specific Manner"

_nutrients, 2023, doi:10.3390/nu15102245_

Round 1
Author Response
Reviewer 1
Based on the Developmental Origin of Health and Disease (DOHaD) concept, the manuscript by Ibanez et al. addresses the depot- and sex-specific effects of maternal obesity in offspring’s adipose tissue. The concept that maternal obesity programs offspring for higher adiposity and metabolic alterations via transcriptional alterations of visceral adipose tissue in a depot- and sex-specific manner in rat is already supported by numerous studies. For example, unlike female, adult male offspring from obese dams was shown to be predisposed to fat accumulation displaying increased visceral (gonadal and perirenal) depots weights, higher serum leptin concentration with elevated lipogenic and diminished adipogenic mRNA levels (Lecoutre et al. J. Endocrinol., 2016).
We agree with the reviewer. There are many studies that support the concept that maternal obesity programs offspring for higher adiposity and metabolic alterations via transcriptional alterations of visceral adipose tissue in a depot- and sex-specific manner in rat. In fact, in the introduction as well as in the discussion, we included our and others references. The article by Lecoutre et al is very interesting, and its inclusion in the discussion will contribute valuable information regarding similarities and differences with our model. The Lecoutre study showed that perirenal adipose tissue is more sensitive to maternal obesity programming than gonadal adipose tissue, and that male offspring of obese mothers are more affected than female offspring.
- One major difference with our study is that we did a comparison of the control group between males and females, the differences allowed us to later analyze the data by sex. Our findings show a clear difference for the control group between males and females in metabolic values, adipocyte size distribution and very importantly gene expression.
- Our work complements Leucotre’s findings. Although the number of genes studied by them (11) and us (84) is limited, the knowledge generated by both groups contributes to a greater understanding of the mechanisms underlying fat accumulation in male and female offspring of obese mothers. In addition, Leucotre et. al. proposes that additional experiments analyzing the lipogenic profile during the development of adipose tissue are required.
- Our study describes an adipogenic expression profile (a crucial aspect of adipose tissue expansion) that includes genes related to adipocyte proliferation and late differentiation. The adipogenic profile identifies genes associated with insulin resistance that are substantially related to one another. Our research suggests that sex-specific insulin resistance in response to maternal obesity may originate predominantly in adipose tissue; however, additional evidence is required to confirm this hypothesis.
In addition, several transcriptomic analysis in rodents have already highlighted the sex-specific effects of maternal obesity on offspring’s adipose tissue (Sava et al. Int J Obes., 2022), in particular the downregulation of insulin pathway linked to worse glycemic control and diabetes predispostion in male vs female.
Thank you very much for the suggestion; the study of Savva and cols is interesting to us. Despite the fact it was conducted on mice rather than rats, it is a different model than ours because the offspring are fed a high fat diet after weaning, whereas we did it with control diet. Therefore, they analyze maternal obesity programming effects after a second hit (high-fat diet in the offspring), while we look at maternal programming without the influence of a high-fat diet in the offspring. However, we consider this paper very important to include in the discussion to highlight that not only maternal diet is important for offspring health, but also the postnatal environment affects the offspring phenotype. Lines 372-378.
Eighty-four adipogenesis-gene mRNA were analyzed in this work but the genes are not listed and the sex-specific difference is too descriptive and not fully exploited.
Based on the reviewer's valuable suggestion, we have added as supplementary material table 1 with all the examined genes and table 2 with all the DEGs F1C female vs. F1C male; F1MO vs. F1C male; and F1MO vs. F1C female (as shown in figure 5). By analyzing 84 genes, we were able to construct Figure 6 showing the functional interaction of DEG, which makes some physiological implications related to the metabolic data. To our knowledge this is new information that contributes to understanding mechanisms involved in adipose tissue programming by maternal obesity. We also have now included more discussion of the signaling pathways affected by sex in C group (lines 445-472) and MO vs C offspring (lines 473-485 and 499-524).
Thus, as stated by the authors, the sex-specific programmation of visceral expansion with cell-size distribution and gene expression modifications are not a hypothesis but an established fact.
We are changing the hypothesis to emphasize the new findings: “considering that the abnormal distribution of AS and its deleterious effects on metabolic function are different between male and female offspring of obese mothers, we hypothesize that gene transcription profiles involved with retroperitoneal adipocyte proliferation and differentiation are sex-specifically regulated”.
Although the paper provides some interesting results, most of them reported here in visceral tissue have already been described in other articles. This study does not bring anything new enough compared to the existing litterature.
We understand reviewer 1's concern; however, table 2 (before table 1) displays the top 10 differentially expressed genes related to adipogenesis by sex in the control group, which to our knowledge have never been reported before in maternal obesity model induced by high-fat diet. Table 2 also displays the differences between MO males and C males in the same 10 genes, as well as the difference between MO females and C females in one single gene. The table highlights: 1) that the differences in gene expression between males and females for the control group show sex dimorphism, thereby justifying the data of MO by sex; 2) that there are differences between females and males patterns of programming by MO, including decreased expression of pro-adipogenic genes and reduced insulin signaling in males and lipid mobilization related genes in females. In addition, our findings explain the decrease in lipogenesis-associated genes found by previous studies and the insulin resistance.
We are changing Table 1, now table 2 incorporating female data for better understanding the differences between male and female programming effects due to maternal obesity. In addition, we are including in the discussion the interpretation of the signaling pathways. We hope that with all these changes, reviewer 1 will be convinced that our work contributes with new knowledge in the area.
In addition, the discussion is too descriptive, and it is a little bit thin on mechanisms. It lacks a consistent discussion on underlying mechanisms regarding sex-specific programming effects of maternal obesity. The authors should have discussed the mechanistic basis for the adipocyte « memory » in their model of maternal obesity in male vs female under the light of 1) developmental window of vulnerability (gestation versus lactation) and its impact on glucose metabolism in offspring later in life and 2) the role of epigenetics in developmental programming.
Thanks for the suggestion, we have now included in the discussion or introduction the adipocyte memory (lines 476-480), the windows of vulnerability (lines 55-61), and the role of epigenetics (lines 476-480).
The authors state that the difference of cellularity in adult female vs male may be due to female estradiol levels. However, in rodents, the perinatal period of life (last week of gestation and lactation) corresponds largely to the period of adipogenesis. In neonates, adipocyte stem cells are highly plastic and very sensitive to maternal factors. In particular, lactation is an active period for hormonal and epigenetic remodelling in offspring before the onset of puberty and sex hormones activities. What are the sex-specific programming mechanisms on metabolic outcomes during the developmental period before puberty?
We appreciate your comment and suggestion, thus we included in the discussion some of the mechanisms regarding sex-specific programming effects of maternal obesity.

Reviewer 2 Report
Maternal overnutrition and maternal obesity is a common phenomena in daily clinical practice and may pose severe long-term risks for the offspring, for example, for the development of obesity and cardiometabolic diseases. The NIH has recently highlighted the importance of differences between sex, and the authors performed a very nice study in this paper. They tested it in a very logical, straightforward way. However, the authors need to consider some factors related to pregnancy and lactation and might also improve some minor things.
Major comments
1. The pups sharing cages with their mums are susceptible to eating the diet before weaning, which usually occurs around 21 PND. How did the authors evaluate it?
2. Food intake in g does not represent the difference in consumption because the HF diet has more caloric density when compared to the control diet. I suggest calculating energy (KJ) density.
3. Are there data on mil production available?
4. Did the authors collect the periovarian fat in females? Once the authors collected epididymal fat, it was expected the males have total fat than females. Explain it.
Minor comments
1. Specify what are F1C and F1MO in the abstract.
2. Why did the authors choose Wistar rats?
3. How was the mated? How many males for females? Please, specify.
Again, specify how the authors did the mated to produce F1.
4. Did the authors use a commercial kit to determine triglyceride and glucose or a protocol? That should be addressed in the methods part 2.5.
5. There are guidelines for formatting gene and protein names; For mice and rats, gene names must be italicised, whit only the first letter in upper-case. Please check the guidelines and change them and all text.
6. Did the authors perform the gene expression using a PCR array? It is not clear in the methods part 2.9. Moreover, add the 84 mRNA transcripts used for the qPCR – it could be as supplementary material.
Author Response
Reviewer 2
Maternal overnutrition and maternal obesity is a common phenomena in daily clinical practice and may pose severe long-term risks for the offspring, for example, for the development of obesity and cardiometabolic diseases. The NIH has recently highlighted the importance of differences between sex, and the authors performed a very nice study in this paper. They tested it in a very logical, straightforward way. However, the authors need to consider some factors related to pregnancy and lactation and might also improve some minor things.
We thank reviewer 2 for his/her kind comment that “The NIH has recently highlighted the importance of differences between sex, and the authors performed a very nice study in this paper”.
Major comments
- The pups sharing cages with their mums are susceptible to eating the diet before weaning, which usually occurs around 21 PND. How did the authors evaluate it?
In our model pups start to eat maternal diet around 11 days of age. Programming effects on the offspring due to the intake of milk and the maternal diet cannot be differentiated from day 11 to 21. To avoid this situation, we studied offspring in young adult offspring (much after weaning) to analyze offspring programming effects rather than diet at that particular moment.
- Food intake in g does not represent the difference in consumption because the HF diet has more caloric density when compared to the control diet. I suggest calculating energy (KJ) density.
We agree with the reviewer, however we are reporting offspring food intake. After weaning, both offspring groups (CF1 and MOF1) were fed with the same diet (control diet). We have previously reported maternal food and calorie intake during pregnancy and found that the calorie intake was similar in both maternal groups (F0C and F0MO). (PMID: 36290594)
- Are there data on milk production available?
We have published changes in milk composition in the maternal obesity model (PMID: 26608475). Milk production and % of water and carbohydrate content were lower in MO compared to C, whereas milk leptin, total fat, AA, monosaturates and fatty acids were higher in MO than in C. Protein content were similar in both groups. We have included this information in the discussion.
- Did the authors collect the periovarian fat in females? Once the authors collected epididymal fat, it was expected the males have total fat than females. Explain it.
We have weighed all fat depots. To answer the reviewer question, we have decided to include the weight of the fat depots (table 1) and explain in the introduction why we studied retroperitoneal adipose tissue. “In rats, the expansion rate of fat depots differs according to their location. Inguinal fat, for instance, exhibits a high degree of hyperplasia, whereas mesenteric and epididymal fat depots exhibit a high degree of hypertrophy. In the present study, retroperitoneal fat tissue was chosen for examination because it is expanded by a balance of hyperplasia and hypertrophy, and its growth is associated with metabolic alterations (PMID: 30524294, 36412162, 7859591, 12351432)”.
Description in results section:
F1C males had increased retroperitoneal, mesenteric, omental and mediastinal fat compared to F1C females. Omental and mediastinal fat were similar in both F1C and F1MO males and females; epididymal fat was similar between male F1C and F1MO; retroperitoneal, parametrial and mesenteric fat depots were similar in female F1C and F1MO. F1MO males had increased retroperitoneal and mesenteric fat depots compared to F1C males and F1MO females increased periovarian fat compared to F1C females (Table 1).
Table 1. Fat depots in male and female F1C and F1MO at 110 PND.
Fat depot (g) |
F1C |
F1MO |
|
Male |
|
Mediastinal |
0.2 ± 0.02 |
0.7 ± 0.3 |
Retroperitoneal |
5.1 ± 0.5 |
8.2 ± 1.4* |
Omental |
0.6 ± 0.1 |
0.7 ± 0.1 |
Mesenteric |
2.5 ± 0.3 |
3.4 ± 0.4* |
Epidydimal |
5.4 ± 0.3 |
6 ± 0.6 |
|
Female |
|
Mediastinal |
0.1 ± 0.02# |
0.3 ± 0.2 |
Retroperitoneal |
2.7 ± 0.2# |
3.1 ± 0.4 |
Omental |
0.4 ± 0.1# |
0.6 ± 0.1 |
Mesenteric |
1.6 ± 0.2# |
2.1 ± 0.3 |
Parametrial |
3.4 ± 0.5 |
4.3 ± 0.6 |
Periovarian |
1.5 ± 0.2 |
2.4 ± 0.4* |
Data are mean ± SEM, n = 8 rats per group, #P< 0.05 vs. F1C male; *P< 0.05 vs. F1C.
Minor comments
- Specify what are F1C and F1MO in the abstract.
We apologize for the omission, we now included in the abstract the meaning of those abbreviations. “…Offspring (F1) from control (F1C) and obese (F1MO) mothers…”
- Why did the authors choose Wistar rats?
The strength of our study is the use of a well-characterized Wistar rat model of maternal obesity. We have studied extensively the current model in Wistar rats (PMID: 30524294, 20351043, 31591717, 32387646, 28063465, 28063465, 34959795, 27496224, 29972240, 26608475, 22239918, 22239918, 34371097, 36072345, 32425146, 34984750, 31179955, 37044135). This strain does not have genetic predisposition for obesity, that helps us to explore the environment effects rather than the interaction of evironment with genes.
- How was the mated? How many males for females? Please, specify.
Again, specify how the authors did the mated to produce F1.
Thank you for the observation, we are incorporating the information you request in the methods section. “Based in the fertility rate from previous studies in this model (PMID: 23949616), at 120 days, 10 female rats from the F0C group and 20 from the F0MO group were mated overnight (up to 5 days) with non-experimental males to generate F1. Daily vaginal smears were obtained, and the day a sperm plug was found was designated as day 0 of conception. F0C and F0MO continued to be fed with C or HFD respectively, throughout pregnancy and lactation (Fig. 1). Fertility rate for F0C was 80% and for F0MO 40%”.
- Did the authors use a commercial kit to determine triglyceride and glucose or a protocol? That should be addressed in the methods part 2.5.
Thank you for the observation, we have included the information in the method section. “Glucose Ref B24985, Cholesterol Ref 467825 and Triglycerides Ref 445850, Beckman Coulter”.
- There are guidelines for formatting gene and protein names; For mice and rats, gene names must be italicised, whit only the first letter in upper-case. Please check the guidelines and change them and all text.
Thanks for the comments, we have checked the guidelines and made the appropriate changes.
- Did the authors perform the gene expression using a PCR array? It is not clear in the methods part 2.9. Moreover, add the 84 mRNA transcripts used for the qPCR – it could be as supplementary material.
Thank you for the suggestion, we will clarify in the methods section that the gene expression was done by PCR array using the RT profile (kit) we are also including the 84 mRNA transcripts in a supplementary table.

Round 2
Reviewer 1 Report
Accepted